# The Role of Biofeedback in Patellofemoral Pain Conservative Treatment: A Systematic Review

**DOI:** 10.3390/jfmk9010021

**Published:** 2024-01-15

**Authors:** Rosario Ferlito, Sara De Salvo, Giovanni Managò, Martina Ilardo, Marco Sapienza, Alessia Caldaci, Andrea Vescio, Vito Pavone, Gianluca Testa

**Affiliations:** 1Department of Biomedical and Biotechnological Sciences, University of Catania, 95123 Catania, Italy; ferlito.rosario@libero.it (R.F.); giovanni_manago@yahoo.it (G.M.); 2Department of General Surgery and Medical Surgical Specialties, Section of Orthopedics and Traumatology, A.O.U. Policlinico-Vittorio Emanuele, University of Catania, Via Santa Sofia 78, 95123 Catania, Italy; sarads94@hotmail.it (S.D.S.); martinailardo52@gmail.com (M.I.); marcosapienza09@yahoo.it (M.S.); alessia.c.92@hotmail.it (A.C.); andreavesscio88@gmail.com (A.V.); gianpavel@hotmail.com (G.T.)

**Keywords:** PFPS, PFP, patellofemoral pain syndrome, patellofemoral pain, anterior knee pain, neurofeedback, biofeedback

## Abstract

This paper aims to investigate the effectiveness and the outcomes of the association between different types of biofeedback techniques and therapeutic exercises in the conservative treatment of patellar femoral pain (PFP). The Preferred Reporting Items for Systematic Reviews and Meta-Analyses (PRISMA) statement guidelines have been used and followed the Cochrane Handbook for Systematic Reviews of Interventions. Between April and June 2023, the following electronic databases were searched: PubMed, ScienceDirect, BIOMED Central, Cochrane Library, and PEDro. Only randomized controlled trials (RCTs) were selected. Following the search, 414 records were found, and after using strict inclusion and exclusion criteria, 12 RCTs were retrieved to include in this systematic review, assessing 513 patients. The association between biofeedback and therapeutic exercise may be beneficial for pain, function (AKPS), extensor muscle strength, reduction of the dynamic knee valgus and vastus medialis (VM) and vastus lateralis (VL) (EMG) optimization. All these results were valued in the short term. Regarding the intervention type, it was possible to correlate the EMG biofeedback with the benefits of the knee extensor strength and the EMG activity of VM and VL. Conversely, using mirror, verbal, and somesthetic (hands and band) feedback seems to be linked to the reduction of the knee dynamic valgus.

## 1. Introduction

Patellofemoral pain syndrome is typical of active young adults, manifesting as retro patellar or peripatellar pain. It usually begins slowly and progresses with a gradual increase in pain. Often atraumatic, it has been linked to increased weight bearing on the patellofemoral joint [1]. The epidemiology of this condition varies widely, mostly due to patients’ fitness; it primarily affects women [2,3].

Many biofeedback methodologies for different diseases have been used in recent years. These techniques give patients information during treatment to influence their neuro-motor functions. These feedback data can be distinguished into visual, auditory, and somesthetic feedback [4].

The literature regarding this is heterogeneous, and we did not find systematic reviews on this topic. We found articles that partially investigate this topic by adding biofeedback to conservative treatment options [5,6,7,8,9,10,11,12,13,14,15], leading us to create this study to analyze, if any, biofeedback’s role in PFPS treatment. 

## 2. Materials and Methods

This systematic review was conducted following the PRISMA guidelines (Preferred Reporting Items for Systematic Reviews and Meta-Analyses) Statement [16] and the Cochrane Handbook for Systematic Reviews of Interventions indications [17].

Inclusion criteria

We included exclusively randomized controlled trials, as they provide the maximum level of evidence, according to directives from the Centre for Evidence-Based Medicine (CEBM) [18,19]. The studies considered are all in English. We did not pose restrictions on the studies’ goals, participant choice, randomization units, number of participants, number of centers involved or assigned treatment knowledge.

2.Participants

We selected studies including PFP patients with different physical activity levels, from sedentary patients to professional athletes. We did not impose limits regarding age, sex, or duration of symptoms. We excluded works focused on other musculoskeletal disorders or other diseases.

3.Types of biofeedback treatments

We considered eligible studies presenting associations between different types of protocols of therapeutic exercise and biofeedback methodologies (visual, auditory, or somesthetic feedback), eventually perfectioned by educational treatments. These studies were compared to those treatments taken singularly.

4.Outcomes

We included studies based on the analysis of the outcomes through scales and validated instruments such as pain (Visual Analogue Scale (VAS), Numeric Pain Rating Scale (NPRS), Patient Specific Functional Scale (PFPS) [20,21], function (AKPS) [22], strength associated to hip abductors muscles and knee extensors, valgus knee with lower extremity cinematics and the electromyography (EMG) of the vastus medialis and lateralis [23,24].

### 2.1. Research Methods

Two revisors (F.P. and G.M.) conducted research between April and June 2022 on the following search motors: PubMed, ScienceDirect, BIOMED Central, Cochrane Library, Google Scholar and PEDro. The words used for the search were: “patellofemoral pain syndrome”, “patellofemoral pain”, “PFPS”, “PFP”, “anterior knee pain”, “patellofemoral chondromalacia”, “chondromalacia patellae”, “exercise therapy”, “motor control”, “motor control exercise”, “exercise”, “sensory-motor training”, “sensorimotor exercise”, “sensorimotor training”, “neurofeedback”, “neurofeedback training”, “proprioception”, “proprioceptive training”, “proprioception training”, “biofeedback”, “visual biofeedback”, “visual feedback”, “proprioceptive exercise”, “proprioceptive feedback”, “neuromuscular training”, “neuromuscular exercise”, and “external focus”.

The combinations were conducted by the Boolean operators (“AND”, “OR” and “NOT”) and, where possible, the terms MeSH (Medical Subject Headings) following this research line according to the Patient, Intervention, Comparison, Outcome (PICO) model [25]. During the research, the potentially relevant studies derived from the bibliographies of the included studies were evaluated.

The used search string is:

((“patellofemoral pain syndrome” [MeSH] OR “patellofemoral pain” OR “PFPS” OR “PFP” OR “patellofemoral pain syndrome” OR “anterior knee pain” OR “patellofemoral chondromalacia” OR “chondromalacia patellae”) AND (“exercise therapy” [MeSH] OR “motor control” OR “motor control exercise” OR “exercise” [MeSH] OR “exercise” OR “sensory-motor training” OR “sensorimotor exercise” OR “sensorimotor training” OR “neurofeedback” OR “neurofeedback training” OR “proprioception” OR “proprioceptive training” OR “proprioception training” OR “biofeedback” OR “visual biofeedback” OR “visual feedback” OR “proprioceptive exercise” OR “proprioceptive feedback” OR “neuromuscular training” OR “neuromuscular exercise” OR “external focus”)).

The search string was modulated on the specific settings of each database.

### 2.2. Study Selection

Our selection process started with duplicate removal by comparing the results in all the databases analyzed. The first selection was successively performed considering the title and abstract of the articles found by the two revisors (M.S. and G.M.). The eventual controversies were resolved with a discussion between these two and a third revisor (R.F.). After all the full texts were found, they were read and analyzed independently by the revisors and selected according to the inclusion criteria. Finally, the eligible articles were evaluated for potential bias risk.

### 2.3. Data Collection, Extraction and Characteristics

We extracted the data from each study from the research topic according to the P.I.C.O. model and the PRISMA statement guidelines [16].

Data extraction was organized according to the following parameters:General information: authors, publishing year, study design;Participants: characteristics, number, gender, age, and PFP duration;Interventions/Controls: participants’ number, type, frequency, and duration;Outcomes;Evaluation and follow-ups;Results: result synthesis with mean and standard deviation (where possible).

### 2.4. Bias Risk Evaluation

Bias risk evaluation was conducted through the Cochrane Collaboration’s Risk of Bias 2 (RoB 2). The RoB2 dominions are:-Bias derived from the randomization process (D1);-Bias due to deviation from the initially planned intervention (D2);-Lack of data bias (D3);-Result measuring bias (D4);-Results report bias (D5);-Overall bias risk (overall).

For every RoB 2 dominion, one of the following judgments was assigned:-Low risk;-Some concerns;-High risk.

### 2.5. Applicability

Following Rothwell’s work [26], an analysis of the following criteria was conducted:-Setting;-Participant selection;-Participant characteristics;-Differences between study protocol and clinical practice;-Outcomes and evaluations;-Adverse events.

### 2.6. Bibliography

The bibliography was completed with Mendeley Reference Management software connected to Microsoft Office Word 2022.

## 3. Results

The research was conducted in six databases, initially finding 414 records. After the removal of 43 duplicates and screening by title and abstract, the remaining records were 24; after full-text reading, 12 RCTs were included in the systematic review, according to the eligibility criteria imposed (flow diagram, Figure 1).

### 3.1. Characteristics of the Studies

Twelve RCTs met the eligibility criteria and were included in the systematic review [27,28,29,30,31,32,33,34,35,36,37,38]. All the studies are in English and published between 2001 and 2021 (see Appendix A).

### 3.2. Dropouts and Lost to Follow-Up

From all RCTs, the dropout total was 11; instead, the number of those lost at follow-up is 6, with a total of 17 participants, as shown in Table 1.

### 3.3. Bias Risk Evaluation

From the evaluation conducted with RoB 2, five studies were found to have a low bias risk [31,32,35,36,37], and four [30,33,34,38] had some concerns. Only three RCTs [27,28,29], the oldest studies, have a high bias risk. The problems encountered in all the studies in the systematic review are mainly about the dominions related to bias derived from the randomization process (D1) and reporting the results (D5). The evaluation of the single dominions of RoB 2 for each study is described through a traffic light plot (Figure 2), and the results are summarized through the summary plot (Figure 3).

### 3.4. Interventions

#### 3.4.1. Effectiveness of Multimodal Interventions

Six studies [27,32,33,34,35,37] present significant heterogeneity in the association between different feedback techniques and therapeutic exercise protocols.

Analyzing the outcomes, we found that in three RCTs [32,35,37], pain was significantly lower in the experimental group when compared to the control. We want to underline that Emamvirdi et al. [35] used a sample as a control group that was subjected only to an educational intervention, with thermotherapy or cryotherapy, while the results for the Roper [32] group are specific to running.

Three other studies [27,33,35] have not found differences in pain between the groups.

In terms of functionality, two RCTs [33,37] came to the same conclusion: the AKPS is significantly better in the experimental group compared to the control at the post-intervention follow-up. During and immediately after the protocol, they found no differences between the groups; this information was also confirmed by Riel et al. [34]. Concerning hip abductor strength, two studies [33,35] did not find differences between the two groups, while Riel et al. [34] found a significant improvement compared to the experimental group. Regarding knee extensors, two studies [34,37] registered an improvement in the intervention group that was significantly better than the controls. In particular, Alonazi et al. [37] underlined that this measure needs at least four weeks before finding a significant difference between the two groups; this underlines the minimum timeframe necessary for this type of adaptation. In contrast, just one study [33] found no differences between the groups. 

The lower limb kinematic was studied in three RCTs [32,33,35], in two of which [32,35] a significative reduction of the knee dynamic valgus in the interventions group related to the single-leg squat (SLS) (Emamviridi et al.) [35] or to the initial contact (IC) for running (Roper et al.) [32], to underline, in this case, the type of control used by the Emamvirdi group [35]. Rabelo and colleagues [33] have not registered any changes regarding the lower limb kinematic, but in this regard, it is essential to underline that the intervention has a relatively limited duration (4 weeks) due to a possible adaptation, as the authors suggest. Finally, only the Dursun group [27] analyzed the EMG activity of vastus medialis (VM) and vastus lateralis (VL), finding a significant improvement in the experimental group if associated with the medium activity of VM and VL; this last study referred only to the first intervention phase (4 weeks) (Table 2).

#### 3.4.2. Effectiveness of the Association between Visual Feedback and Therapeutic Exercise

From the analysis of four studies [28,29,30,38] that evaluated the effectiveness of the association between visual feedback and therapeutic exercise, we found that the outcome pain studied in two works [28,38] was reduced significantly in both groups, considering the RCT of Yip and colleagues [28]. However, there are no significant differences regarding the decrease between the two; instead, in the Ebrahimi group’s work [38], a significative reduction exclusively in the case group was underlined. However, it is essential to note that the intervention used Kinect for visual feedback instead of the EMG biofeedback of the other studies, and the controls did not receive a therapeutic exercise protocol but an educational one. These last researchers [38] were the only group to focus on functionality, measured through AKPS, underlining, in this case, a significative improvement in favor of the intervention group, with the above limitations. Only the two RCTs of Bennell [30] and Yip [28] et al. evaluated knee extensor strength: the first found no significant differences in strength improvement in the two groups, except for the eccentric strength immediately after the end of the protocol (6 weeks) in favor of the interventions protocol. The Yip [28] group found a significantly higher improvement in this variable in the case group immediately after the intervention (8 weeks) but an improvement in the controls at the evaluation intra-intervention (4 weeks). Lastly, the analysis regarding the EMG of the quadriceps, conducted by two studies [29,30], found that regarding the activation of VMO and VL, there was a significant improvement in the interventions group, as underlined by Ng et al. [29]. However, if we consider the VMO-VL EMG onset timing studied by Bennell et al. [30], this had a significant improvement in favor of the intervention group exclusively for stair descents and immediately successive to the end of the intervention (6 weeks). All the details are reported in Table 3.

#### 3.4.3. Efficacy of the Association between Auditory Feedback and Therapeutic Exercise

Only one RCT included in this study [31] focused on the effects of the association of auditory feedback and therapeutic exercise. The data showed that, regarding pain reduction, the intervention group protocol showed significantly better efficacy compared to the control at the follow-up (12 weeks). Regarding strength, a significant improvement in the experimental group on the hip abductors was registered, which is foreseeable considering the differences in the therapeutic protocol; instead, regarding knee extensors, they have not encountered any significant differences from the baseline in both groups. The data that differs from other studies is that they found a significant reduction of efficacy related to the intervention group on knee abduction. The above is summarized in Table 4.

#### 3.4.4. Efficacy of the Association between Somesthetic Feedback and Therapeutic Exercise

Only one study [36] investigated the effects of the association between somesthetic feedback and therapeutic exercise by comparing three groups. It appears that the external focus (EF) group has better outcomes, all significantly improved if compared to the control group; instead, only the knee dynamic valgus looks significantly diminished compared to the internal focus (IF) group. See Table 5.

## 4. Discussion

### 4.1. Biofeedback and Therapeutic Exercise Association on Pain

Evaluating the effectiveness of the association between different biofeedback methodologies and therapeutic exercise protocols concerning pain, it seems that just two studies [32,37] with a low bias risk underlined a significantly better improvement in favor of the experimental groups that followed multimodal interventions (mirror and verbal feedback or EMG biofeedback and taping); this does not let us discriminate which feedback type significantly impacts the result. The other three RCTs [31,35,38] found actual effectiveness in this intervention. However, the type of control groups used does not let us draw any conclusion because Baldon et al. (2014) [31] used different therapeutic protocols between interventions and control groups and found a significant improvement only after the follow-up (12 weeks). Emamvirdi (2019) [35] and Ebrahimi (2021) [38] accounted only for the controls with an educational treatment. Instead, Aghakeshizadeh et al. (2021) [36] found a significant improvement in the case group compared to the control group treated only with therapeutic exercise and did not find differences if compared to a third group with a different protocol (exercise and internal focus) if compared to the experimental protocol (exercise and external focus). However, four studies [27,28,33,34] did not find differences between the two groups analyzed, even if they presented high bias risk.

### 4.2. Biofeedback and Therapeutic Exercise Association on Functionality

The efficacy of functionality was measured through the AKPS, and results were significant in only two of the works we considered [33,37]. These conclude that the improvement in the score of the Kujala Scale is significantly in favor of the experimental group only if the variable evaluated in the follow-up in the short term (6 or 12 weeks) is considered. In comparison, there are no differences between the groups if done intra-intervention (2 weeks), post-intervention (4 weeks), or middle-term follow-up. In addition, in this case, though, being the proposal multimodal (mirror, verbal and proprioceptive feedback or EMG biofeedback and taping), it is impossible to discriminate which protocol type is most relevant to the outcome. As highlighted for pain, Ebrahimi et al. (2021) [38] noticed a significant improvement for the experimental group also on functionality (AKPS), but with the same “problems” mentioned above in the control group. Aghakeshizadeh et al. (2021) [36], similar to the outcome pain, registered a significant improvement in the AKPS for the intervention group to the control one performing exclusively exercise, while compared with the third group, assigned to a different protocol, did not find differences. Finally, only Riel et al. (2018) [34] did not encounter significative differences in the groups, but this work does have some bias risk.

### 4.3. Biofeedback and Therapeutic Exercise Association on Strength

#### 4.3.1. Hip Abductors

Only one study [34] showed a significant improvement in efficacy in favor of the intervention group on the hip abductors muscle strength, underlining that an association between visual and auditory feedback with therapeutic exercise protocol can favor strength improvement in these areas if compared to exercise only. In addition, Aghakeshizadeh et al. (2021) [36] found some improvements but, similar to pain and functionality outcomes, these were found only in the comparison between the exercise and external focus group compared to the control group treated with therapeutic exercise, which was non-significant when compared to the group with exercise and internal focus. The RCT of Baldon et al. (2014) [31] presented critical findings because there was a significant improvement in hip abductors’ strength related to the experimental group, but it must be noted that this group was doing concrete work on these muscle districts. On the other side, only Rabelo and colleagues (2017) [33] did not find differences between the two groups analyzed, considering the bias risk. Strangely, Emamviridi et al. (2019) [35] did not find any differences between the groups analyzed, even if the control one did not exercise; this can be explained by the fact that the duration of the protocol (6 weeks) was not sufficient to register adaptations of this type, and also by the nature of the intervention itself.

#### 4.3.2. Knee Extensors

Regarding knee extensors, three RCTs [28,34,37] found a significant improvement in favor of the experimental group regarding this variable, even with a high bias risk. Two works [28,37] found that the strength improvement was verified only after the intervention (4 or 8 weeks) and at the following eventual follow-up (6 weeks). In comparison, there were no differences between the groups at the evaluation intra-intervention (2 or 4 weeks), underlining the necessity of a longer protocol duration to obtain a positive adaptation. Bennell et al. (2010) [30] found a significant improvement in favor of the intervention group related to the eccentric strength evaluated just after the intervention (6 weeks), but was canceled at the following follow-up (14 weeks); instead, in terms of the concentric and isometric strength, there were no found differences between the two groups at post-intervention nor follow-up. Moreover, all the studies that reported knee extensor strength improvement used visual feedback, especially the EMG biofeedback, in the protocols proposed for the interventions. Finally, in the studies run by Baldon (2014) [31] and Rabelo (2017) [33], even if this former had some bias risk, they did not find differences between the groups studied.

### 4.4. Biofeedback and Therapeutic Exercise Association on Lower Limb Kinematic

Aghakeshizadeh et al. (2021) [36] reached concrete conclusions regarding the lower limb kinematic of the knee valgus, underlining significant improvements in favor of the experimental group (exercise and external focus) compared to both in the control group, with just exercise, and the group doing a protocol with the association of exercise and internal focus. This work presents a low bias risk; therefore, we may suppose that an association between somesthetic feedback (operator’s hands and band) and therapeutic feedback can be beneficial to reduce knee dynamic valgus. Roper’s study (2016) [32], with a low bias risk, also underlines a reduction of the knee valgus related to IC while running, which was significative only in the experimental group and exclusively after the follow-up (4 weeks); this underlines that the intervention length (2 weeks) is not sufficient to obtain this adaptation. Significant improvement related only to the intervention group was encountered by Emamvirdi et al. (2019) [35], even when a patient education protocol was administered to the control group, and the RCT may have had some biases. Both studies use an intervention protocol composed of therapeutic exercise, mirror and verbal feedback. Rabelo et al. (2017) [33] did not find differences in the examined groups, but the short duration of the intervention (4 weeks) is a limiting factor in the development of this adaptation; moreover, this study presents some biases. Finally, only the RCT conducted by Baldon (2014) [31] found a reduction in knee abduction during the SLS in their experimental group in contradiction with the other studies; while this last study presented a low risk of bias, we underline the differences of the therapeutic exercise protocols between the experimental and control groups.

### 4.5. Biofeedback and Therapeutic Exercise Association on VM and VL EMG Activity

In the included studies, different aspects relating to the influence of the intervention on the VM (vastus medialis) and VL (vastus lateralis) EMG activity of the quadriceps muscle were discussed. Dursun et al. (2001) [27] registered significant improvements in the VM medium EMG activity in favor of the experimental group during all the recordings. Regarding VL, they found that the medium EMG activity increased in favor of the intervention group only with the evaluation intra-intervention (4 weeks); instead, no differences were registered in the remaining groups (8 and 12 weeks). Any differences between the analyzed groups regarding the maximum VM and VL EMG activity were described. However, it should be highlighted that this study presents a high bias risk. Ng et al.’s (2008) [29] RCT, presenting a high bias risk, found a significative optimization of the VMO/VL EMG ratio in favor of the experimental group. Finally, the study done by Bennell et al. (2010) [30] showed a significant improvement in the intervention group if compared to the control one, exclusively associated with the VMO-VL EMG onset timing in going down the stairs during the evaluation post-intervention (6 weeks) that disappeared at the following follow up (12 weeks), no differences were described in this outcome related to ascending the stairs in both the evaluations. The analysis of this study’s results had to consider the possible presence of bias. It is important to underline how the cited studies associated therapeutic exercise with EMG-biofeedback with eventual auditory feedback in the experimental group. This combination can represent a valuable aid to optimize VM and VL EMG activity.

## 5. Applicability

We found some elements that might limit the results’ applicability regarding external validity. First, there are differences in the setting used, from clinic rooms [27,33,37] to research laboratories [31,33,35,38]; it is relevant to underline that some researchers [28,29,30,32,34,36] have not given any specific information about this. Another essential thing to notice is the inclusion and exclusion criteria, which are very selective and heterogeneous, even if all the patients analyzed are patients with PFP. Regarding the operators that were doing the evaluations and supervising the interventions, even if they were mainly physiotherapists or doctors, some studies underline the extreme experience and specialization of these last ones [30,31,33,36,37] and the deepening of the interventions mentioned above [32,33,35,38]. Moreover, it is hard to replicate the use of specific instrumentations through which the feedback component of the intervention is conducted (EMG-biofeedback [27,28,29,30,37] or BandCizer61).

Moreover, most of the evaluations and follow-ups were conducted in the short term (<3 months), and this offers an opportunity to know the duration of the possible adaptations induced by the protocols. Furthermore, even though some of the studies [28,33,34,38] reported no adverse events and only one RCT [32] signaled the rising of temporary symptoms like pain in the experimental group, we underline that many studies [27,29,35,36,37] did not provide any information about this matter. Considering the dropouts and those lost to follow-up, we want to emphasize that the reasons for these results were not due to the nature of the intervention but to the impossibility of contacting the patient [30,31,33,34], personal reasons [30,31,36,38] and testing positive for SARS-CoV-2 [38].

## 6. Limitations

This study has several limitations. A limiting factor might be represented by the type of interventions and controls used because they present a heterogeneous nature between each other, which is hard to interpret for the use of multimodal interventions, for which it is not clear which ones had more impact on the outcome. Moreover, some outcomes were evaluated with different systems, complicating the confrontation between the studies. Finally, we want to note that we also considered studies with high bias risk, but this can be a limitation because it might impact the RCT results included in this systematic review.

## 7. Conclusions

It is impossible to underline definitive conclusions about biofeedback use in the conservative treatment of PFP. It aids the patients in terms of pain control and restoring function.

Regarding the benefit of biofeedback in association with therapeutic exercise on strength, we want to underline that, for hip abductors, the data does not show significant benefits because even if there are some studies in its favor, they have biases.

Regarding knee extensors, the evidence is more robust than that of hip abductors, even if some studies do not find any significant differences. In particular, this treatment needs a long intervention length (6–8 weeks), and its efficacy decreases if it is not prolonged in time (14 weeks). Moreover, in all the RCTs that found benefits related to this outcome, visual feedback was associated with therapeutic exercise, particularly EMG biofeedback, which might be related to this adaptation.

Concerning the influence of visual feedback and therapeutic exercise on the kinematics of the lower extremity, there were significant improvements in the valgus knee with an adequate intervention duration (6–8 weeks). It seems that all feedback types might be helpful to obtain this result, in particular, mirror feedback, verbal feedback, and somesthetic feedback, such as the use of professional hands or bands.

Finally, through a protocol using therapeutic exercise and EMG biofeedback (with eventual auditory feedback), there were significant changes associated with VM and VL activity, in particular with an increase in the EMG mean activity for VL after a brief treatment duration (4 weeks), an increase in VMO-VL EMG onset timing during descending the stairs in the short term post-intervention (6 weeks) and an improvement of VMO/VL EMG ratio after treatment (8 weeks).

### Implications for Clinical Practice

From the conclusions, we can underline that biofeedback methodologies might be an exciting means to include in therapeutic protocols for PFP, but considering the heterogeneity of the interventions and the results, as well as the criticality of the applicability of some protocols, it might be advisable to adopt only the feedback types that are compliant to clinical practice, of easy use, available and economical like mirror feedback or verbal feedback, or the help of external foci such as hands or bands to obtain a possible added benefit to the outcomes that show significative positive effects (pain, function, knee dynamic valgus).

## Figures and Tables

**Figure 1 jfmk-09-00021-f001:**
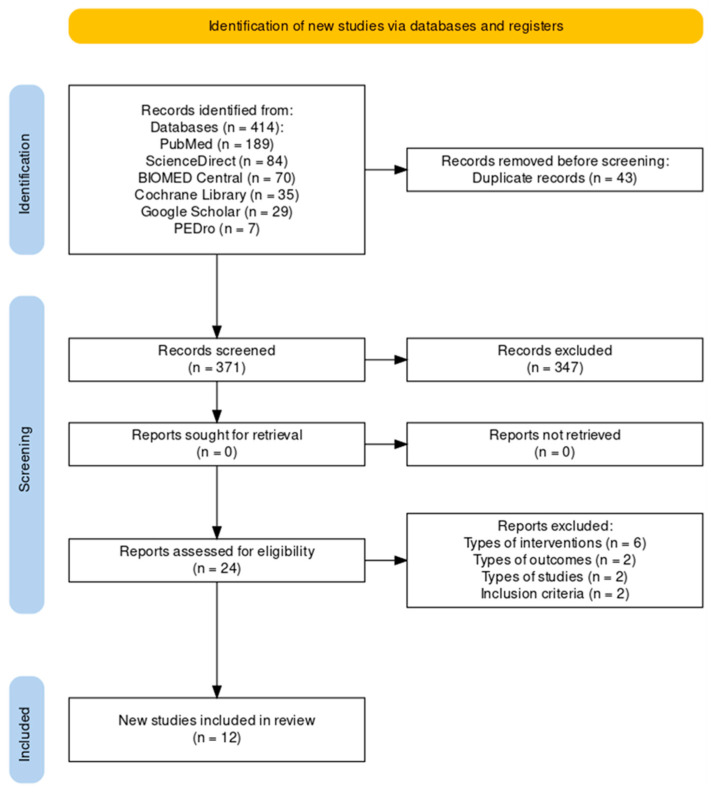
Flow diagram.

**Figure 2 jfmk-09-00021-f002:**
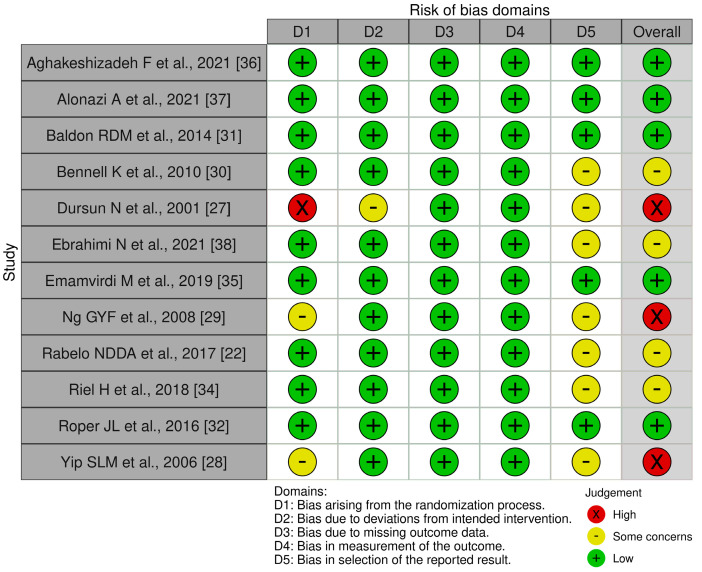
Traffic light plot.

**Figure 3 jfmk-09-00021-f003:**
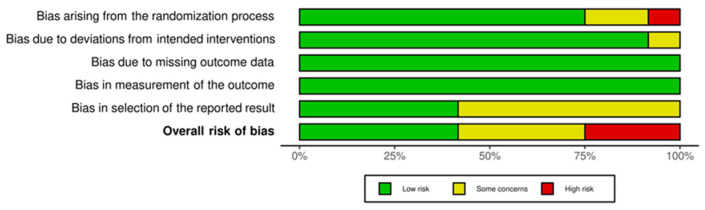
Summary plot.

**Table 1 jfmk-09-00021-t001:** Dropouts and lost to follow-up.

	Drop-Outs		Losts to Follow-Up	
Studio	Intervention	Control	Intervention	Control
Aghakeshizadeh F et al., 2021 [36]	2 (EF)	2 (IF); 1 (C)	-	-
Alonazi A et al., 2021 [37]	0	0	0	0
Baldon RDM et al., 2014 [31]	0	1	0	1
Bennell K et al., 2010 [30]	0	1	3	0
Dursun N et al., 2001 [27]	0	0	-	-
Ebrahimi N et al., 2021 [38]	1	1	-	-
Emamvirdi M et al., 2019 [35]	0	0	-	-
Ng GYF et al., 2008 [29]	0	0	-	-
Rabelo NDDA et al., 2017 [33]	0	0	1	1
Riel H et al., 2018 [34]	1	1	-	-
Roper JL et al., 2016 [32]	0	0	0	0
Yip SLM et al., 2006 [28]	0	0	-	-

**Table 2 jfmk-09-00021-t002:** Multimodal intervention effectiveness.

Study	Intervention	←	No Difference between the Groups	→	Control
Alonazi A et al., 2021 [37]	Exercise + EMG-BF + Patellar taping (McConnell)	VAS (p_week2_ = 0.0008; p_week4,6_ = 0.0005) AKPS (p_week6_ = 0.0002) Knee strength EXT (p_week4,6_ = 0.0008)	AKPS (p_week2_ = 0.086; p_week4_ = 0.171) Strength EXT knee (p_week2_ = 0.259)		Exercise + EMG-BF false + Taping placebo
Dursun N et al., 2001 [27]	Exercise (strengthening − stretching − bike) + EMG-BF + Auditory FB	Activity EMG VM (medium) (p_week4_ = 0.046; p_week8_ = 0.042; p_week12_ = 0.036) Activity EMG VL (medium) (p_week4_ = 0.007)	Activity EMG VM (maximum) (p_week4_ = 0.283; p_week8_ = 0.1; p_week12_ = 0.13) Activity EMG VL (maximum) (p_week4_ = 0.267; p_week8_ = 0.061; p_week12_ = 0.099) Activity EMG VL (medium) (p_week8_ = 0.052; p_week12_ = 0.14) VAS (p_week4_ = 0.149; p_week8_ = 0.532; p_week12_ = 0.14)		Exercise (strengthening − stretching − bike)
Emamvirdi M et al., 2019 [35]	Exercise + Verbal and mirror FB	VAS (*p* = 0.001) Knee valgus reduction SLS (*p* = 0.004)	ABD hip strength (*p* = 0.127)		Education (posture instructions and suggestions for the overall health) + Heat/Ice pack (1–2/weeks)
Rabelo NDDA et al., 2017 [33]	Pre-exercise education (on disorders of motor control) Exercise + Proprioceptive (unstable surfaces), verbal and mirror FB	AKPS (p_week12_ = 0.04)	NPRS (p_week4,12,24_ > 0.05) AKPS (p_week4,24_ > 0.05) Hip ABD strength (*p* > 0.05) Knee EXT strength (*p* > 0.05) Knee ADD SDT (*p* > 0.05)		Exercise
Riel H et al., 2018 [34]	Exercise + Visual and auditory FB (BandCizer, TUT and pulling force)	Hip ABD and knee EXT (*p* = 0.048)	AKPS (*p* = 0.28) VAS (*p* > 0.05)		Exercise + Visual FB (BandCizer, pulling force)
Roper JL et al., 2016 [32]	Education (written indications) + Gait retraining + Mirror and verbal FB	Knee ABD (IC running) (p_sett4_ < 0.05) VAS (running) (p_sett2,4_ < 0.05)	Knee ABD (IC running) (p_sett2_ > 0.05)		Gait training + Mirror FB

**←**—Favorable effect for the intervention group; **→**—favorable effect for the control group; EMG-BF—electromyography biofeedback; FB—feedback; VAS—Visual Analogue Scale; NPRS—Numeric Pain Rating Scale; AKPS—Kujala Anterior Knee Pain Scale; ABD—abductors/abduction; ADD—adduction; EXT—extensors; SLS—single-leg squat; SDT—step-down test; IC—initial contact; EMG—electromyography; VM—vastus medialis; VL—vastus lateralis.

**Table 3 jfmk-09-00021-t003:** Effectiveness of the association between visual feedback and therapeutic exercise.

Study	Intervention	←	No Difference between the Groups	→	Control
Bennell K et al., 2010 [30]	Exercise + EMG-BF	VMO-VL EMG onset (descending stairs) (p_sett6_ = 0.02) Knee EXT eccentric strength (p_sett6_ = 0.004)	VMO-VL EMG onset (going up the stairs) (p_sett6_ > 0.05; p_sett14_ = 0.85) VMO-VL EMG onset (descending stairs) (p_sett14_ = 0.81) Knee EXT eccentric strength (p_sett14_ = 0.14) Knee EXT concentric strength (p_sett6_ = 0.18; p_sett14_ = 0.99) Knee EXT isometric strength (p_sett6_ = 0.06; p_sett14_ = 0.76)		Exercise
Ebrahimi N et al., 2021 [38]	Education (on training and ADL) + Exercise + Visual FB (Kinect)	VAS (*p* = 0.004) AKPS (*p* < 0.001)			Education (on training and ADL)
Ng GYF et al., 2008 [29]	Exercise + EMG-BF	VMO/VL EMG ratio (*p* = 0.016)			Exercise
Yip SLM et al., 2006 [28]	Exercise (stretching − strengthening − balance and proprioception − plyometric and agility training) + EMG-BF	Knee EXT strength (p_sett8_ = 0.023)	PFPS Severity Scale (p_sett4,8_ = 0.088)	Knee EXT strength (p_sett4_ = 0.032)	Exercise (stretching − strengthening − balance and proprioception − plyometric and agility training)

**←**—Favorable effect for the intervention group; **→**—favorable effect for the control group; EMG-BF—electromyographic biofeedback; FB—feedback; VAS—Visual Analogue Scale; AKPS—Kujala Anterior Knee Pain Scale; ABD—abductors; EXT—extensors; EMG—electromyography; VMO—vastus medialis oblique; VL—vastus lateralis.

**Table 4 jfmk-09-00021-t004:** Effectiveness of the association between auditory feedback and therapeutic exercise.

Study	Intervention	←	No Difference between the Groups	→	Control
Baldon RDM et al., 2014 [31]	Quadriceps strengthening − hip muscles − core + Verbal FB	VAS (p_sett12_ = 0.04) Knee ABD reduction SLS (*p* = 0.004) Hip ABD strength (*p* = 0.001)	VAS (p_sett8_ = 0.06) Knee EXT strength (*p* > 0.05)		Quadriceps strengthening + Stretching

**←**—favorable effect for the intervention group; **→**—favorable effect for the control group; FB—feedback; VAS—Visual Analogue Scale; ABD—abductors/abduction; EXT—extensors; SLS—single-leg squat.

**Table 5 jfmk-09-00021-t005:** Effectiveness of the association between somesthetic feedback and therapeutic exercise.

Study	Intervention	←	No Difference between the Groups	→	Control
Aghakeshizadeh F et al., 2021 [36]	EF group Exercise + EF (hands and band)	VAS (p_C_ = 0.02) AKPS (p_C_ = 0.03) Hip ABD strength (p_C_ = 0.01) Knee valgus reduction SLS (p_C_ = 0.01; p_IF_ = 0.03)	VAS (p_IF_ = 0.27) AKPS (p_IF_ > 0.05) Hip ABD strength (p_IF_ > 0.05)		IF group Exercise + IF (thinking about adjustments) C group Exercise

**←**—Favorable effect for the intervention group; **→**—favorable effect for the control group; EF—external focus; IF—internal focus; C—control; VAS—Visual Analogue Scale; AKPS—Kujala Anterior Knee Pain Scale; ABD—abductors; SLS—single-leg squat.

## Data Availability

All data are available in the manuscript.

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
