# Peer review of "The Role of Biofeedback in Patellofemoral Pain Conservative Treatment: A Systematic Review"

_jfmk, 2024, doi:10.3390/jfmk9010021_

Round 1

Reviewer 1 Report

Comments and Suggestions for Authors

Author Response

Dear reviewer thank you for the time spent for our systematic review.

We make a lot of changes, following your evaluable suggestions.

We hope you will appreciate numerous changes performed. We are proud to consider further suggestions to improve the article, if you retain necessary.

Thank you again

Reviewer 2 Report

Comments and Suggestions for Authors

Thank you for the opportunity to review this manuscript. It presents a very careful revision of the literature on a troubling topic. However, I have some comments and suggestions presented below.

2. Materials & Methods

Line 52. Please explain the acronym CEBM. 

Line 59. Please explain the acronym MSK. 

Line 61. Please explain a little more the “types of biofeedback treatments”. For example, in the last sentence “These studies were compared to those treatments taken alone”, it is not easily understood to which treatments you refer to. 

Line 66. Please explain the acronyms in all the paragraph. 

Line 84. Please explain the acronym PICO

3. Results

Figure 1. In the reports excluded: inclusion criteria n=2, to which criteria do you refer exactly? For example, type of study has been included previously. 

Please, include in the legend of the Table in Appendix A the explanation of the acronyms used. Please add the number of the reference in column 1 of the table in Appendix A. 

Line 173. Please revise the syntax. We do not understand to which associations do you refer to. Do you mean variability in the feedback techniques included in the exercise protocols?  

Please add the number of the reference in column 1 of the Tables 2, 3 y 4.  

Line 193. Please correct the style of the reference 33. 

Lines 195-203. Please add the explanation of the acronyms in the text. I know that in this case you have included them in the legend of the table, but the explanation must also appear the first time they are included in the text. 

Lies 254-259. Please, explain the acronyms EF and IF. 

7. Conclusion

Please, add some words at the start of the conclusions regarding the effects on pain and functionality. 

Comments on the Quality of English Language

 Moderate editing of English language required

Author Response

Dear Reviewer thank you for your evaluable considerations and suggestions

Here you will find a point-by-point revision:

Materials and Methods

Q1) Line 52. Please explain the acronymus CEBM

A1) Done

Q2) Line 59. Please explain the acronym MSK.

A2) Changed in musculoskeletal

Q3) Line 61. Please explain a little more the “types of biofeedback treatments”. For example, in the last sentence “These studies were compared to those treatments taken alone”, it is not easily understood to which treatments you refer to.

A3) the sentence has been modified

Q4) Line 66. Please explain the acronyms in all the paragraph.

A4) All acronyms have been explained

Q5) Line 84. Please explain the acronym PICO

A5) The acronym has been explained

 3.⁠ ⁠Results

Q6) Figure 1. In the reports excluded: inclusion criteria n=2, to which criteria do you refer exactly? For example, type of study has been included previously.

A6) The inclusion criteria of those 2 studies were not complete and did not meet the estabished criteria of our systematic review

Q7) Please, include in the legend of the Table in Appendix A the explanation of the acronyms used. Please add the number of the reference in column 1 of the table in Appendix A.

A7) Done

Q8) Line 173. Please revise the syntax. We do not understand to which associations do you refer to. Do you mean variability in the feedback techniques included in the exercise protocols? 

A8) We mean the variability of feedback techniques. The phrase has been modified

Q9) Please add the number of the reference in column 1 of the Tables 2, 3 y 4.

A9)  Done

Q10) Line 193. Please correct the style of the reference 33.

A10) Done

Q11) Lines 195-203. Please add the explanation of the acronyms in the text. I know that in this case you have included them in the legend of the table, but the explanation must also appear the first time they are included in the text.

A11) Done

Q12) Lies 254-259. Please, explain the acronyms EF and IF.

A12)  Done

 7.⁠ ⁠Conclusion

Q13) Please, add some words at the start of the conclusions regarding the effects on pain and functionality.

A13) A sentence has been added

Round 2

Reviewer 1 Report

Comments and Suggestions for Authors

Nothing to add.